# Intraoperative alteration in the vital signs of diabetic patients during cataract surgery with local anesthesia

Yuka Kojima, Norihiko Misawa, Tatsunori Yamamoto, Shigeru Honda ᵒ *

Department of Ophthalmology and Visual Sciences, Osaka City University Graduate School of Medicine, Osaka, Japan

* honda.shigeru@med.osaka-cu.ac.jp

## Abstract

### Purpose

Diabetic patients often have systemic circulation diseases which may cause serious systemic complications during ophthalmic surgeries with local anesthesia. The purpose of this study is to evaluate the intraoperative alteration of the vital signs in diabetic patients during cataract surgery with local anesthesia.

### Methods

Clinical records of 428 patients who underwent cataract surgeries with local anesthesia were reviewed. The parameters measured were systolic/diastolic blood pressures and pulse rates at pre-operation, 5, 10 and 15 minutes during the surgeries. The factors were compared between non-diabetic patients (n = 325) and diabetic patients (n = 103).

### Results

Diabetic patients had significantly higher fasting blood sugar and preoperative pulse rate. Diabetic patients showed significantly higher systolic blood pressure compared to non-diabetic patients at 5 and 10 minutes from the beginning of surgery (p = 0.0093 and 0.0075, respectively). In the non-diabetic patients, the pulse rate was significantly decreased at 5 minutes from the beginning of surgery (p = $4.74 \times 10^{-8}$) which was maintained during the surgery, but no change was observed in the pulse rate of the diabetic patients.

### Conclusions

Diabetic patients showed higher systolic blood pressure and pulse rate during cataract surgeries with local anesthesia, which should be monitored carefully by the physicians to avoid possible systemic complications.

**Data Availability Statement:** All relevant data are within the manuscript and its Supporting Information files.

**Funding:** The authors received no specific funding for this work.

**Competing interests:** The authors have declared that no competing interests exist.

## Introduction

Diabetes mellitus (DM) is a prevalent disorder in the developed countries which causes several complications in the systemic circulation including hypertension, cardiac infarction and strokes [1]. Cardiovascular autonomic neuropathy is frequently observed in patients with DM. As anesthesia has a marked effect on peri-operative autonomic function, the interplay between diabetic neuropathy and anesthesia may result in unexpected hemodynamic instability during surgery [2]. Those complications may occur during ophthalmic surgeries performed under local anesthesia, which may cause serious problems in the ocular and systemic conditions [3]. To investigate the risk factors of intraoperative abnormal vital signs when performing cataract surgeries with local anesthesia is an important issue for ophthalmologists. On the other hand, a recent technical advancement of cataract surgery enables safer and faster operation which expands the indication of cataract surgery with local anesthesia to the patients with more systemic risk factors [4, 5]. However, few studies have been reported regarding the hemodynamic state of DM patients during a recent minimal invasive cataract surgery with local anesthesia [6].

Here, we conducted a statistical evaluation on the intraoperative alteration of the vital signs of DM patients during cataract surgery with local anesthesia.

## Materials and methods

This study is a retrospective chart review approved by the Institutional Review Board at the Osaka City University Graduate School of Medicine (No. 2020–153) and was conducted in accordance with the Declaration of Helsinki. All cases in this study were Japanese individuals recruited from the Department of Ophthalmology at Osaka City University Hospital in Japan. Written informed consent for using ordinary clinical data in following retrospective studies were obtained from all subjects at their first visit to the hospital and an opt-out for this study was indicated at the hospital website.

Clinical records of 428 patients who underwent the initial cataract surgeries with local anesthesia between June 1st 2018 and May 31th 2019 were reviewed after anonymization of personal information.

The standard procedure of cataract surgery in the present study was as follows: 1) All patients were advised to take all their usual medications with a small amount of water on the morning of surgery. 2) Abstinence from solid food 2 hours preoperatively. 3) Pupil dilation with 10% phenylephrine and 1% tropicamide eyedrops. 4) Topical anesthesia with 0.4% oxybuprocaine was administered twice preoperatively with 5-minute intervals between instillations. 5) A certified nurse inserted a 22-gauge caliber after a venous puncture in the arm in either upper extremity, depending on vein availability. 6) The parameters monitored intraoperatively included continuous pulse rate, pulse oximetry, electrocardiography, and blood pressure every 5 minutes recorded. 7) Several topical drops of 4% lidocaine hydrochloride and a sub-Tenon's block with 0.5–1.0 ml of 2% lidocaine hydrochloride were applied to the operated eye. 8) A 2.4 mm self-sealing sclerocorneal incision was made and phacoemulsification + aspiration was performed. 9) A foldable posterior chamber intraocular lens was implanted. In the present study, all cataract surgeries were performed by experienced surgeons. Cases with any complications, such as early perforation, capsule rupture or zonular disinsertion were excluded from the study. The surgeries required operation time for more than 50 minutes were also excluded.

Baseline parameters of the patients collected were age, gender, body weight, body mass index (BMI), fasting blood sugar (FBS) at the latest visit, estimated glomerular filtration rate (eGFR), medications for DM or systemic hypertension. In the present study, the patients

receiving any medication for DM were determined as diabetic patients. Ones receiving only diet therapy were not determined as diabetic patients. The parameters measured were systolic/diastolic blood pressures (SBP/DBP) and pulse rates (PR) at pre-operation, 5, 10 and 15 minutes during the surgeries. The factors were compared between non-diabetic patients (n = 325) and diabetic patients (n = 103).

For statistical analysis, each clinical factor was evaluated by paired or unpaired *t*-test where applicable between any two groups. To find the independent factors associated with the vital signs during the surgery multiple regression analyses were performed with explanatory variables including age, gender, BMI, FBS, eGFR, pre-operative SBP/DBP and PR, the presence or absence of DM and the presence or absence of hypertension medication.

We used EZR1.27 software for all the statistical analyses [7]. P values of 0.05 or less were considered to be statistically significant.

## Results

Clinical characteristics of the patients with or without DM are summarized in Table 1. The patients with DM showed significantly higher BMI, higher FBS, higher pre-operative pulse rate and higher proportion of patients receiving hypertension medication compared to the patients without DM.

The mean SBPs of DM and non-DM patients significantly increased at 5 minutes ($9.50 \times 10^{-15}$ and $3.23 \times 10^{-49}$, respectively) and gradually decreased over time. The mean SBPs of DM patients was significantly higher than those of non-DM patients at 5 and 10 minutes from the beginning of surgery (Fig 1). The mean DBPs in DM and non-DM groups significantly increased at 5 minutes ($9.64 \times 10^{-16}$ and $5.27 \times 10^{-39}$, respectively), which sustained up to 15 minutes from the beginning of surgery (Fig 2). There was no difference in the mean DBPs between two groups at any period measured. The mean PRs of DM patients was significantly

**Table 1. Clinical characteristics of the patients with or without DM.**

|  | DM- (n = 325) | DM+ (n = 103) | P-value |
|---|---|---|---|
| Gender (male/female) | 143/182 | 49/54 | 0.60 * |
| Age (mean±SD) | 75.3±10.4 | 73.1±9.4 | 0.057 † |
| BMI | 23.2±3.7 | 24.1±3.9 | 0.047 |
| FBS (mg/dl) (mean±SD) | 103.2±18.0 | 148.6±71.3 | $4.62 \times 10^{-23}$ † |
| eGFR (mL/min/1.73m$^2$) (mean±SD) | 61.3±17.7 | 58.0±21.2 | 0.12 |
| Pre-operative SBP (mmHg) (mean±SD) | 128.1±19.9 | 131.5±19.0 | 0.12 † |
| Pre-operative DSBP (mmHg) (mean±SD) | 70.7±13.2 | 68.7±11.4 | 0.16 † |
| Pre-operative PR (beats/min) (mean±SD) | 69.5±11.8 | 72.7±12.0 | 0.017 † |
| Hypertension Medication (yes/no) (%) | 197/128 (60.6) | 79/24 (76.7) | 0.0043 * |
| Operation time (minutes) (mean±SD) | 21.5±8.8 | 22.4±9.8 | 0.35 |

DM; diabetes mellitus, BMI; body mass index, FBS; fasting blood sugar, eGFR; estimated glomerular filtration rate, SBP; systolic blood pressure, DSBP; diastolic blood pressure, PR; pulse rate.

* $\chi^2$ test

† unpaired *t*-test. Values are indicated as mean ± standard deviation where applicable.

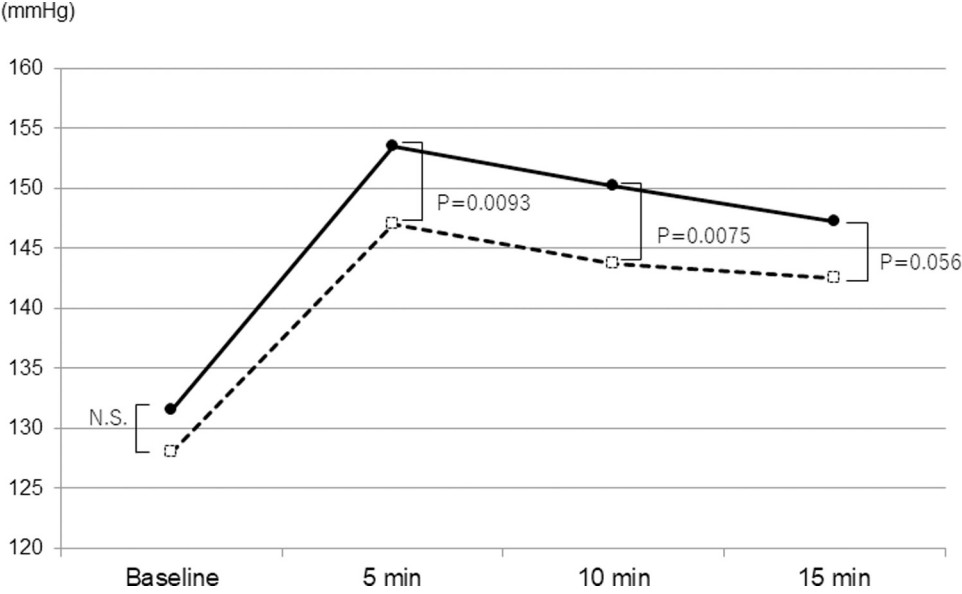

**Fig 1. Time course of systolic blood pressure in the DM group (solid line) and non-DM group (dashed line).** P-values are the results of unpaired *t*-test.

higher than those of non-DM patients at all time point measured (Fig 3). A significant decrease in the mean PR was observed at 5 minutes from the beginning of surgery in non-DM patients, but not in DM patients.

The multiple regression analyses demonstrated that the significant association factors for SBP were FBS and eGFR at 5 and 10 minutes from the beginning of surgery (Table 2), and the

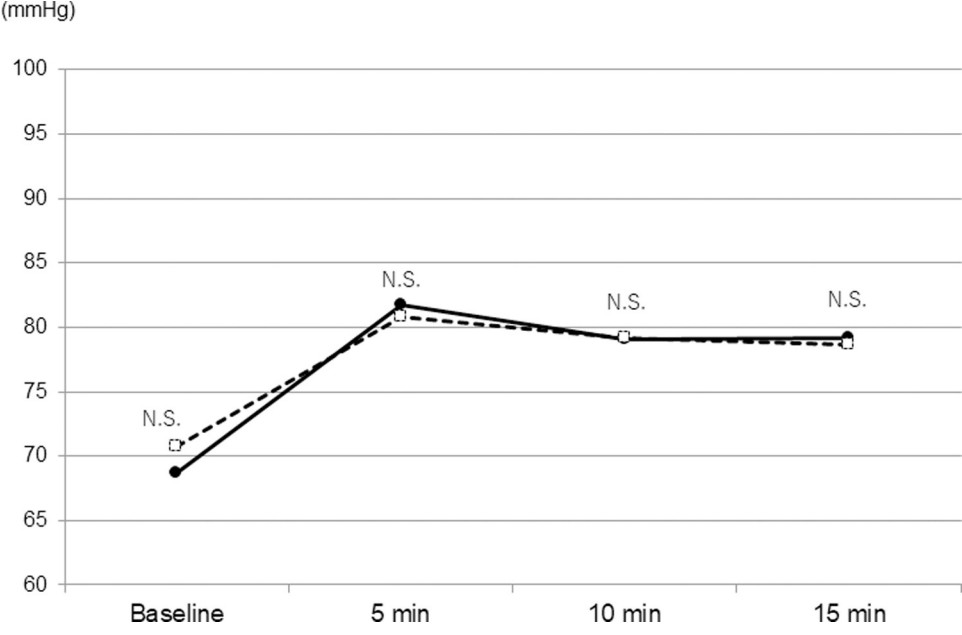

**Fig 2. Time course of diastolic blood pressure in the DM group (solid line) and non-DM group (dashed line).** P-values are the results of unpaired *t*-test.

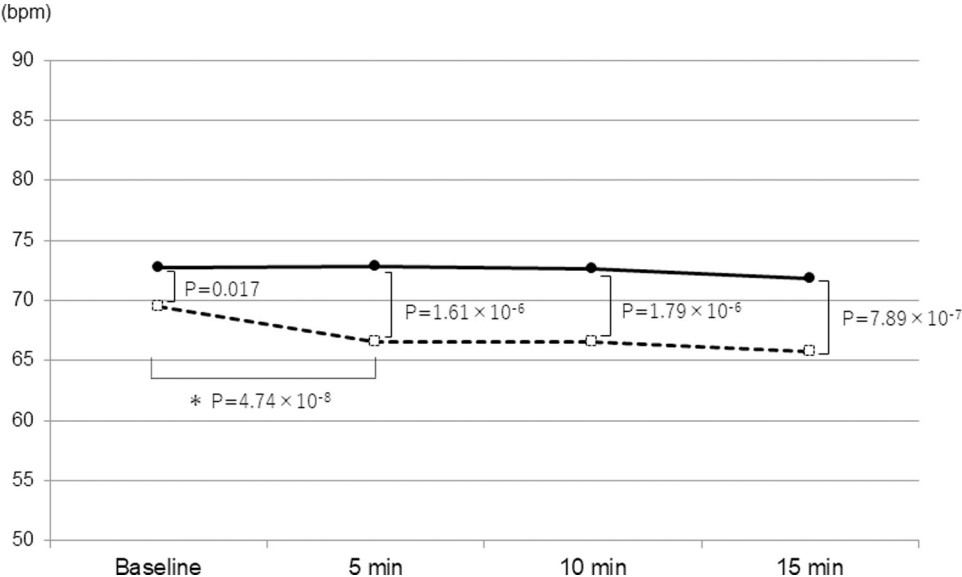

**Fig 3. Time course of pulse rate in the DM group (solid line) and non-DM group (dashed line).** P-values are the results of unpaired or paired (*) *t*-test.

significant association factor for PR was the presence of DM at 5, 10 and 15 minutes from the beginning of surgery (Table 3). FBS was also associated with the PR at 5 minutes.

## Discussion

We evaluated the intraoperative alteration of vital signs in DM patients during cataract surgery with local anesthesia and demonstrated that DM patients showed significantly higher SBP and PR than non-DM patients during cataract surgery with local anesthesia.

In the present day, cataract surgery is established with a fast and low invasive procedure which enables many diabetic patients to undergo cataract surgeries without serious ocular and systemic complications [4, 5]. However, an anxiety for the surgery may still affect some physical or mental conditions of the patients during the operation with local anesthesia [3, 8], hence we must pay attention to any changes in the vital signs of the patients during the surgery. In particular, diabetic patients often exhibit systemic hypertension and following cardiovascular

**Table 2. Results of stepwise multiple regression analysis for a systolic blood pressure.** Explanatory variables including age, gender, BMI, FBS, eGFR, pre-operative SBP/DBP and PR, the presence or absence of DM and the presence or absence of hypertension medication.

| | Estimate | SE | *t* value | *P* value |
|---|---|---|---|---|
| A. Significant association factors at 5 minutes. | | | | |
| FBS | 0.12 | 0.024 | 5.05 | $6.70\times10^{-7}$ |
| eGFR | -0.20 | 0.055 | -3.59 | $3.64\times10^{-4}$ |
| B. Significant association factors at 10 minutes. | | | | |
| | Estimate | SE | *t* value | *P* value |
| FBS | 0.085 | 0.024 | 3.61 | $3.38\times10^{-4}$ |
| eGFR | -0.20 | 0.054 | -3.74 | $2.10\times10^{-4}$ |

DM; diabetes mellitus, BMI; body mass index, FBS; fasting blood sugar, eGFR; estimated glomerular filtration rate, SBP; systolic blood pressure, DSBP; diastolic blood pressure, PR; pulse rate, SE; standard error.

**Table 3. Results of stepwise multiple regression analysis for a pulse rate.** Explanatory variables including age, gender, BMI, FBS, eGFR, pre-operative SBP/DBP and PR, the presence or absence of DM and the presence or absence of hypertension medication.

| | Estimate | SE | t value | P value |
|---|---|---|---|---|
| A. Significant association factors at 5 minutes. | | | | |
| FBS | 0.032 | 0.014 | 2.19 | $2.93 \times 10^{-2}$ |
| DM+ | 4.63 | 1.44 | 3.21 | $1.41 \times 10^{-3}$ |
| B. Significant association factors at 10 minutes. | | | | |
| DM+ | 5.85 | 1.26 | 4.65 | $4.36 \times 10^{-6}$ |
| C. Significant association factors at 15 minutes. | | | | |
| DM+ | 5.95 | 1.21 | 4.91 | $1.28 \times 10^{-6}$ |

DM; diabetes mellitus, BMI; body mass index, FBS; fasting blood sugar, eGFR; estimated glomerular filtration rate, SBP; systolic blood pressure, DSBP; diastolic blood pressure, PR; pulse rate, SE; standard error, DM+; presence of diabetes mellitus.

diseases which may cause serious prognosis, which necessitates more careful control of blood pressure and pulse rate during the surgery [9]. In the present study, the mean baseline SBP/DBP in DM group was not significantly different from that of non-DM group probably because many patients received medications for systemic hypertension, which could influence the results. However, the mean SBP in DM group was more than 150 mmHg at 5–10 minutes from the beginning of surgery, which was significantly higher than that of non-DM group. Some recent reports recommended to target SBP/DBP at 130-140/80-90 mmHg in adults with DM and arterial hypertension in order to reduce the risk of lethal events [10–12]. Hence, our results indicate the need of more careful monitoring and control of SBP during cataract surgery in DM patients. The multiple regression analyses revealed that SBP was significantly associated with FBS and eGFR which was consistent with previous reports [6, 13, 14]. Hence, the higher SBP in DM patients may reflect the abnormal renal function and glucose metabolism. In addition, the mean pulse rate in DM group was significantly higher at all time point measured in this study. Interestingly, the mean pulse rate in non-DM group was significantly decreased after 5 minutes from the beginning of surgery which was not observed in DM group. A previous report demonstrated that DM patients have an alteration of the autonomic nervous system which is clearly more marked under mental stress than in resting condition [9]. We considered that a temporal elevation of PR occurred preoperatively with an anxiety for the surgery with local anesthesia which was attenuated over time with autonomic control in non-DM patients, but this function did not work in DM patients. This hypothesis was supported with the results of multiple regression analyses which determined the presence of DM as the only significant factor associated with the perioperative PR.

The limitation of the present study is mostly related to the retrospective study design which might be affected by undefined confounding factors. For example, the PR and SBP could be affected by the pain sensation during the surgery, which was not evaluated in the present study. A hypotensive drug might be injected for the patients who showed sustained high SBP during the surgery depends on the surgeons' discretions, which could influence the results of the present study. A systematic prospective study may warrant the results of this study. However, our results indicated that DM patients could show higher SBP and PR than non-DM patients during the latest cataract surgery. Therefore, we still need to be more careful with the vital signs of DM patients during cataract surgeries with local anesthesia.

## Supporting information

**S1 Data.**
(XLSX)

## Author Contributions

**Conceptualization:** Shigeru Honda.

**Data curation:** Yuka Kojima, Norihiko Misawa, Tatsunori Yamamoto.

**Formal analysis:** Yuka Kojima, Shigeru Honda.

**Investigation:** Yuka Kojima, Norihiko Misawa, Tatsunori Yamamoto.

**Methodology:** Yuka Kojima, Norihiko Misawa.

**Resources:** Norihiko Misawa.

**Supervision:** Tatsunori Yamamoto, Shigeru Honda.

**Validation:** Norihiko Misawa, Tatsunori Yamamoto, Shigeru Honda.

**Writing – original draft:** Yuka Kojima.

**Writing – review & editing:** Tatsunori Yamamoto, Shigeru Honda.

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
