## [Decision Letter · Decision Letter 0]

31 Jan 2022

PONE-D-21-22264

Intraoperative alteration in the vital signs of diabetic patients during cataract surgery with local anesthesia.

PLOS ONE

Dear Dr. Honda,

Thank you for submitting your manuscript to PLOS ONE. After careful consideration, we feel that it has merit but does not fully meet PLOS ONE’s publication criteria as it currently stands. Therefore, we invite you to submit a revised version of the manuscript that addresses the points raised during the review process.

We look forward to receiving your revised manuscript.

Kind regards,

Suho Lim

Academic Editor

PLOS ONE

Journal Requirements:

Additional Editor Comments (if provided):

Dear Shigeru Honda

Manuscript entitled PONE-D-21-22264 "Intraoperative alteration in the vital signs of diabetic patients during cataract surgery with local anesthesia." which you submitted to PLOS ONE, has been reviewed.

It has been considered by our two reviewers who found it to be interest, however extensive revisions are required for it to be suitable for publication.

Please refer to the review comments listed carefully. Especially, the reviewers raised significant concerns and would like to address these issues. Please pay careful attention to each of the points raised by reviewers.

Thank you for your contribution.

Sincerely yours.

Reviewers' comments:

Reviewer's Responses to Questions

**Comments to the Author**

1. Is the manuscript technically sound, and do the data support the conclusions?

Reviewer #1: Yes

Reviewer #2: Yes

2. Has the statistical analysis been performed appropriately and rigorously? 

Reviewer #1: Yes

Reviewer #2: Yes

3. Have the authors made all data underlying the findings in their manuscript fully available?

Reviewer #1: No

Reviewer #2: Yes

4. Is the manuscript presented in an intelligible fashion and written in standard English?

Reviewer #1: Yes

Reviewer #2: Yes

5. Review Comments to the Author

Reviewer #1: This study evaluated the intraoperative vital signs and associated factors under local anesthesia. I propose several points to be clarified.

1. The authors described the operation was performed within 30 mins. Please provide the mean operation time in table 1.

2. Were there any patients who needed additional medications due to high blood pressure before the surgery? If so, please describe that point in method section, and provide that information in result section such as the proportion of patients who were administered additional medication to control the high BP.

3. Please provide the ocular information affecting the cataract surgery. Were all cases senile cataract? And if it is possible, axial length, presence of uveitis, or previous surgical history should be provided.

4. The pulse rate and systolic BP can be affected by the pain sensation due to surgical procedure such as iris touching. The pupil of DM patients usually dilated poorly. If it is possible, please state the information about the pupil size or posterior synechia.

5. In the same context as point 4, young patients usually feel pain better than older patients. Although, the mean age of two groups is not different significantly, DM patients were younger in table 1. I think presenting the factors which is not significant (including age) in table 2 and 3 would be helpful for better understanding of readers.

Reviewer #2: The authors evaluated the intraoperative alteration of the vital signs in diabetic patients during cataract surgery with local anesthesia and emphasized the need for careful monitoring by the physicians to avoid possible systemic complications However, there are several concerns to be clarified before publication.

1. In figure 3, I think it is necessary to correct the figure legend as "Time course of pulse rate"

2. In figure 1, the mean SBP in DM group was more than 150 mmHg at 5-10 minutes from the beginning of surgery. Did you take any treatment for that?

3. In general, cataract surgery does not cause much pain, but blood pressure and pulse rate could be different depending on the severity of pain. It is better to describe whether you have considered this.

4. There were several reports that bradycardia, hypotension associated lidocaine toxicity. Therefore, the possibility that lidocaine administration used for sub-tenon’s block may affect purse rate or blood pressure should be considered.

6. PLOS authors have the option to publish the peer review history of their article (what does this mean?). If published, this will include your full peer review and any attached files.

Reviewer #1: No

Reviewer #2: **Yes: **Areum Jeong

---

## [Author Response · Author response to Decision Letter 0]

2 Feb 2022

Author response to the reviewers

We greatly appreciate the reviewers for many valuable comments given to our manuscript. We would provide the point-by-point answers for each comment. In addition, we would announce that the number of DM group has been changed from 104 to 103 after recounting the subjects and values of associated parameters have been slightly changed after recalculation, which does not affect the conclusion.

Reviewer #1: This study evaluated the intraoperative vital signs and associated factors under local anesthesia. I propose several points to be clarified.

1. The authors described the operation was performed within 30 mins. Please provide the mean operation time in table 1.

Response: Thank you very much for this comment. Firstly, we apologize for a mistyping of operation time. Actually, all surgeries were performed within 50 minutes (not 30 minutes) which has been corrected in the text (page 6, line 9). The mean operation times have been added in Table 1. In addition, vales of BMI and eGFR have also been presented in Table 1.

2. Were there any patients who needed additional medications due to high blood pressure before the surgery? If so, please describe that point in method section, and provide that information in result section such as the proportion of patients who were administered additional medication to control the high BP.

Response: We do not use additional hypotensive drugs before cataract surgery and rarely use such drugs during the surgery. If systolic BP of a patient is over 200 mmHg we firstly advise the patient to relax and repeat BP measurements several times (usually every 5 minutes). If BP is not decreased after repeated BP measurements, an injection of hypotensive drug might be used depends on surgeon’s discretion. Hence, it is quite rare to use additional hypotensive drugs within 15 minutes from the beginning of surgeries. However, we have mentioned about the possible use of hypotensive drugs as a limitation of this study in discussion part. (page 15, line 4-6)

3. Please provide the ocular information affecting the cataract surgery. Were all cases senile cataract? And if it is possible, axial length, presence of uveitis, or previous surgical history should be provided.

Response: Thank you for this comment. We agree that ocular status may affect cataract surgery and difficult procedures might cause intraoperative pain which could influence vital signs. However, in the present study, we obtained the data after anonymization of personal information (patient’s name, ID of clinical record etc.) and no correspondence table was made. Hence, we cannot obtain additional data about ocular status or past history.

4. The pulse rate and systolic BP can be affected by the pain sensation due to surgical procedure such as iris touching. The pupil of DM patients usually dilated poorly. If it is possible, please state the information about the pupil size or posterior synechia.

Response: We really appreciate the reviewer for this advice since we did not include intraoperative pain as a possible association factor with vital signs. Unfortunately, we cannot search additional information from anonymized data. Therefore, we have mentioned the possible influence of intraoperative pain as a limitation of this study in discussion part. (page 15, line 2-4)

5. In the same context as point 4, young patients usually feel pain better than older patients. Although, the mean age of two groups is not different significantly, DM patients were younger in table 1. I think presenting the factors which is not significant (including age) in table 2 and 3 would be helpful for better understanding of readers.

Response: We have presented all explanatory variables in Table 2 and 3 which should help the readers to understand non-significant factors.

Reviewer #2: The authors evaluated the intraoperative alteration of the vital signs in diabetic patients during cataract surgery with local anesthesia and emphasized the need for careful monitoring by the physicians to avoid possible systemic complications However, there are several concerns to be clarified before publication.

1. In figure 3, I think it is necessary to correct the figure legend as "Time course of pulse rate"

Response: Thank you very much for this comment. We have corrected the mistyping.

2. In figure 1, the mean SBP in DM group was more than 150 mmHg at 5-10 minutes from the beginning of surgery. Did you take any treatment for that?

Response: We rarely use additional hypotensive drugs during cataract surgery. If systolic BP of a patient is over 200 mmHg we firstly advise the patient to relax and repeat BP measurements several times (usually every 5 minutes). If BP is not decreased after repeated BP measurements, an injection of hypotensive drug might be used depends on surgeon’s discretion. Hence, it is quite rare to use additional hypotensive drugs within 15 minutes from the beginning of surgeries. However, we have mentioned about the possible use of hypotensive drugs as a limitation of this study in discussion part. (page 15, line 4-6)

3. In general, cataract surgery does not cause much pain, but blood pressure and pulse rate could be different depending on the severity of pain. It is better to describe whether you have considered this.

Response: We appreciate the reviewer for this advice. Actually, we did not evaluate intraoperative pain in this study. However, unfortunately, we cannot perform further evaluation for additional factors since the original data of present study were obtained after anonymization of personal information (patient’s name, ID of clinical record etc.) without correspondence table. Therefore, we have mentioned the possible influence of intraoperative pain as a limitation of this study in discussion part. (page 15, line 2-4)

4. There were several reports that bradycardia, hypotension associated lidocaine toxicity. Therefore, the possibility that lidocaine administration used for sub-tenon’s block may affect purse rate or blood pressure should be considered.

Response: Thank you for this comment. We agree that lidocaine may affect vital signs. In the present study, sub-Tenon’s lidocaine was administrated for both of DM patients and non-DM patients, so we consider that the difference between these two groups would remain under the possible influence of lidocaine.

Revised parts are indicated in red. We hope that all reviewers’ comments have been adequately addressed. Again, thank you so much for reviewing our manuscript.

---

## [Decision Letter · Decision Letter 1]

24 Feb 2022

Intraoperative alteration in the vital signs of diabetic patients during cataract surgery with local anesthesia.

PONE-D-21-22264R1

Dear Dr. Honda,

We’re pleased to inform you that your manuscript has been judged scientifically suitable for publication and will be formally accepted for publication once it meets all outstanding technical requirements.

Kind regards,

Suho Lim

Guest Editor

PLOS ONE

Additional Editor Comments (optional):

Reviewers' comments:

Reviewer's Responses to Questions

**Comments to the Author**

1. If the authors have adequately addressed your comments raised in a previous round of review and you feel that this manuscript is now acceptable for publication, you may indicate that here to bypass the “Comments to the Author” section, enter your conflict of interest statement in the “Confidential to Editor” section, and submit your "Accept" recommendation.

Reviewer #1: All comments have been addressed

Reviewer #2: All comments have been addressed

2. Is the manuscript technically sound, and do the data support the conclusions?

Reviewer #1: (No Response)

Reviewer #2: Yes

3. Has the statistical analysis been performed appropriately and rigorously? 

Reviewer #1: (No Response)

Reviewer #2: Yes

4. Have the authors made all data underlying the findings in their manuscript fully available?

Reviewer #1: (No Response)

Reviewer #2: Yes

5. Is the manuscript presented in an intelligible fashion and written in standard English?

Reviewer #1: (No Response)

Reviewer #2: Yes

6. Review Comments to the Author

Reviewer #1: The manuscript has been well improved. I think most of the comments have been addressed. In my opinion this paper is worth to be published in Plos one.

Reviewer #2: (No Response)

7. PLOS authors have the option to publish the peer review history of their article (what does this mean?). If published, this will include your full peer review and any attached files.

Reviewer #1: No

Reviewer #2: No

---

## [Editor Report · Acceptance letter]

14 Mar 2022

PONE-D-21-22264R1 

Intraoperative alteration in the vital signs of diabetic patients during cataract surgery with local anesthesia. 

Dear Dr. Honda:

I'm pleased to inform you that your manuscript has been deemed suitable for publication in PLOS ONE. Congratulations! Your manuscript is now with our production department. 

Kind regards, 

on behalf of

Dr. Suho Lim 

Guest Editor

PLOS ONE